# Interplay of Ferroptosis, Cuproptosis, Autophagy and Pyroptosis in Male Infertility: Molecular Crossroads and Therapeutic Opportunities

**DOI:** 10.3390/ijms26083496

**Published:** 2025-04-08

**Authors:** Difan Cai, Junda Li, Zekang Peng, Rong Fu, Chuyang Chen, Feihong Liu, Yiwang Li, Yanjing Su, Chunyun Li, Wei Chen

**Affiliations:** Health Science Center, Hunan Normal University, Changsha 410013, China; 202230191095@hunnu.edu.cn (D.C.); 202230191103@hunnu.edu.cn (J.L.); 202330191135@hunnu.edu.cn (Z.P.); 202330191154@hunnu.edu.cn (R.F.); chreden@hunnu.edu.cn (C.C.); feihongliu2024@hunnu.edu.cn (F.L.); 202330191245@hunnu.edu.cn (Y.L.); 202030191120@hunnu.edu.cn (Y.S.); lichunyun@hunnu.edu.cn (C.L.)

**Keywords:** male infertility, ferroptosis, cuproptosis, autophagy, pyroptosis

## Abstract

Male infertility is intricately linked to dysregulated cell death pathways, including ferroptosis, cuproptosis, pyroptosis, and autophagy. Ferroptosis, driven by iron-dependent lipid peroxidation through the Fenton reaction and inactivation of the GPX4/Nrf2/SLC7A11 axis, disrupts spermatogenesis under conditions of oxidative stress, environmental toxin exposure, or metabolic disorders. Similarly, cuproptosis—characterized by mitochondrial dysfunction and disulfide stress due to copper overload—exacerbates germ cell apoptosis via FDX1 activation and NADPH depletion. Pyroptosis, mediated by the NLRP3 inflammasome and gasdermin D, amplifies testicular inflammation and germ cell loss via IL-1β/IL-18 release, particularly in response to environmental insults. Autophagy maintains testicular homeostasis by clearing damaged organelles and proteins; however, its dysregulation impairs sperm maturation and compromises blood–testis barrier integrity. These pathways intersect through shared regulators; reactive oxygen species and mTOR modulate the autophagy–pyroptosis balance, while Nrf2 and FDX1 bridge ferroptosis–cuproptosis crosstalk. Therapeutic interventions targeting these mechanisms have shown promise in preclinical models. However, challenges persist, including the tissue-specific roles of gasdermin isoforms, off-target effects of pharmacological inhibitors, and transgenerational epigenetic impacts of environmental toxins. This review synthesizes current molecular insights into the cell death pathways implicated in male infertility, emphasizing their interplay and translational potential for restoring spermatogenic function.

## 1. Introduction

Male infertility is an increasingly recognized public health concern, with a global infertility rate of approximately 17.5% [1]. Systematic reviews and meta-analyses of retrospective cohort studies have demonstrated a significant decline in sperm counts among men worldwide from 1973 to 2018, encompassing regions including North America, Europe, Australia, South and Central America, Africa, and Asia [2]. Beyond its impact on reproductive health, severe male infertility is associated with an increased risk of chronic conditions, including cardiovascular diseases, autoimmune disorders, and malignant tumors such as testicular and prostate cancer [3]. Additionally, reduced sperm counts and concentrations have been associated with an increase in mortality rate [4,5]. A prospective cohort study of 384,419 Danish men revealed that infertile individuals exhibited a higher overall mortality rate compared to their fertile counterparts [6]. Furthermore, mortality risk demonstrates a dose-dependent relationship with the severity of semen quality impairment [7]. Therefore, male infertility constitutes a critical public health challenge, with significant implications for both individual reproductive health and population dynamics. Male infertility is multifactorial, influenced by environmental pollutants (e.g., endocrine disruptors), genetic anomalies (e.g., Y-chromosome microdeletions), congenital abnormalities (e.g., cryptorchidism), infections (e.g., SARS-CoV-2-induced pediatric acute epididymo-orchitis), lifestyle factors (e.g., smoking, obesity), and sociodemographic trends such as delayed parenthood [8,9,10,11,12].

The most prevalent etiology of male infertility, accounting for approximately 75% of cases, is impaired spermatogenesis secondary to primary testicular failure [13]. Spermatogenesis is a tightly regulated biological process wherein spermatogonial stem cells undergo self-renewal, differentiate through spermatocyte stages into spermatozoa, and ultimately mature into functionally competent sperm. This intricate process relies on a precisely coordinated testicular microenvironment comprising Sertoli cells, peritubular myoid cells, and Leydig cells, which collectively synthesize and secrete cytokines critical for spermatogenesis. Dysregulation of any cellular component with this microenvironment can lead to spermatogenic failure, consequently manifesting as male infertility [10]. Emerging evidence implicates regulated cell death pathways in the pathogenesis of male infertility (Figure 1). Distinct molecular mechanisms—including ferroptosis, cuproptosis, autophagy, and pyroptosis—have been identified as key mediators of germ cell demise, with activation patterns influenced by both the pathological state and environmental exposure. This review systematically examines the mechanistic interplay between these cell death modalities and male reproductive dysfunction, with particular emphasis on their pathophysiological significance and translational potential.

## 2. Ferroptosis

### 2.1. Male Infertility Is Closely Related to Ferroptosis

Ferroptosis, a form of cell death mediated by iron metabolism and lipid peroxidation, has been implicated in the pathogenesis of male infertility and impaired spermatogenesis. Clinical studies on male infertility or related comorbidities have shown abnormal levels of ferroptosis markers. Bioinformatic and machine learning studies suggest that the downregulation of heat shock transcription factor 1 (HSF1), Kelch domain-containing 3 (KLHDC3), and glutathione peroxidase 4 (GPX4), which is a selenium-dependent antioxidant enzyme critical for neutralizing lipid peroxides, thereby preventing ferroptosis, may be involved in ferroptosis in non-obstructive azoospermia (NOA) [14]. In semen samples from asthenozoospermic patients, the levels of GPX4 and solute carrier family 7 member 11 (SLC7A11), which is a cystine/glutamate antiporter essential for glutathione synthesis, acting as a gatekeeper of ferroptosis susceptibility, were found to be lower than those in normozoospermic individuals, while reactive oxygen species (ROS), malondialdehyde (MDA), and iron concentrations were significantly elevated. A strong positive correlation between Gpx4 mRNA levels and sperm motility was observed, suggesting that reduced GPX4 expression may be a key factor in the ferroptosis mechanism underlying male infertility [15]. Varicocele (VC), a prevalent cause of male infertility, is characterized by abnormal dilation of the pampiniform venous plexus, leading to impaired thermoregulation, oxidative stress (e.g., reduced glutathione peroxidase activity), and compromised spermatogenesis [16,17]. Furthermore, in diabetic patients, sperm motility is decreased, while ROS and lipid peroxidation are increased. Both type 1 and type 2 diabetes mellitus lead to low ejaculation volume and decreased sperm motility [18]. Additionally, hepatitis infections can cause sperm defects, and exposure to hepatitis B surface antigens can downregulate GPX4 expression in Sertoli cells, indicating that hepatitis may induce male infertility via ferroptosis in Sertoli cells [19].

Ferroptosis also plays a role in male infertility induced by environmental pollutants (Figure 2). Studies indicate that exposure to particulate matter(PM) such as PM2.5 and PM10, cigarette smoke, cadmium (Cd), arsenite, and other environmental stressors [20,21,22,23,24,25,26,27,28,29,30,31,32] triggers ferroptosis in testicular cells, especially in Sertoli cells [23,28,33], Leydig cells [22], and germ cells [20,21,24]. Heavy metals, such as Cd, significantly impact nuclear factor erythroid 2-related factor 2 (Nrf2). Nrf2 has been identified as a master regulator of antioxidant defense, activating GPX4 to counteract oxidative stress. Beyond ferroptosis, Nrf2 dysregulation is implicated in neurodegenerative diseases, cancer, and metabolic disorders, underscoring its broad pathophysiological relevance [34]. Cd exposure increases Nrf2 expression and activates its downstream signaling pathways. By contrast, zinc (Zn) counteracts Cd-induced damage by increasing metabolically active Sertoli cells and alleviating Sertoli cell injury, partially through *Nrf2* regulation [35]. Hydrogen sulfide also prevents ferroptosis in Sertoli cells by suppressing intracellular acrolein accumulation [33]. This is associated with ROS accumulation, disruption of iron homeostasis, and subsequent activation of ferroptosis. The molecular mechanisms involve the upregulation of ferroptosis-related genes and proteins, such as ferredoxin 1 (FDX1), which has been identified as a mitochondrial iron-sulfur protein implicated in cuproptosis, linking copper toxicity to lipoylated tricarboxylic acid (TCA) cycle enzyme aggregation [36] and the downregulation of antioxidant defenses, leading to lipid peroxidation and cell death [22,27,28,29,37]. Antioxidants or ferroptosis inhibitors have shown potential in mitigating testicular toxicity and improving sperm quality, making ferroptosis inhibition a promising strategy for treating male infertility caused by environmental exposure [21,24,33]. Related randomised controlled trials suggest antioxidant supplementation may improve live birth rates in subfertile males [38].

Furthermore, ferroptosis inhibitors or natural extracts may improve male infertility by suppressing ferroptosis. Ferroptosis inhibitors mainly exert their effects by inhibiting iron accumulation and lipid peroxidation [39]. Ferrostatin-1 (Fer-1) and deferoxamine (DFO), two well-characterized ferroptosis inhibitors, mitigate oxidative stress and lipid peroxidation in testicular cells [40,41]. In busulfan-induced oligospermia models, Fer-1 restored sperm quality by upregulating Gpx4 and modulating Nrf2 signaling, underscoring its therapeutic potential in male infertility [40,41]. Similarly, compounds such as ursonic acid [42], aucubin [43], cordycepin [44], myristic acid [45], hirsutella sinensis [46], mangiferi [47], and Semen Cuscutae and Fructus Lycii [48] have demonstrated protective effects against spermatogenic damage by modulating ferroptosis and enhancing antioxidant defenses, highlighting their potential therapeutic value in treating male infertility. These findings underscore the importance of targeting ferroptosis as a therapeutic approach for male infertility.

### 2.2. Mechanisms of Ferroptosis in Male Infertility

Key factors in the ferroptosis pathway mediate male infertility through the following mechanisms. ROS, byproducts of aerobic metabolism, include superoxide anions (O_2_^•^^−^), hydroxyl radicals (•OH), hydrogen peroxide (H_2_O_2_), and singlet oxygen (^1^O_2_). In the presence of mitochondrial superoxide dismutase (SOD, EC 1.15.1.1), O_2_^•^^−^ is converted to H_2_O_2_, which then diffuses from mitochondria into the cytoplasm. At high iron concentrations that favor the Fenton reaction, H_2_O_2_ generates highly reactive O_2_^•^^−^ groups, and further peroxide accumulation induces ferroptosis in germ cells [49]. Imbalances in ROS production and antioxidant defenses are key contributors to male infertility. ROS and calcium (Ca^2+^) homeostasis disturbances activate mitochondrial permeability transition pores (mPTP), leading to mitochondrial dysfunction in sperm [50]. Studies in infertility models have shown that elevated peroxides levels and excessive ROS accumulation promote lipid peroxidation and subsequent ferroptosis [22,24,29,51]. Collectively, ROS accumulation is a major driver of ferroptosis-mediated male infertility.

Glutathione (GSH) is a tripeptide thiol compound synthesized via cysteine transport into cells through the system Xc^−^ glutamate antiporter. GSH exerts antioxidant effects by scavenging lipid peroxides, reducing ROS production, and inhibiting ferroptosis [52]. Glutathione synthase (GSS, EC 6.3.2.3) catalyzes the final step in GSH synthesis. In *GSS*-deficient cells, compromised antioxidant capacity leads to lipid peroxidation and ferroptosis. This deficiency downregulates GPX4 and upregulates arachidonic acid 15 lipoxygenase (ALOX15), a downstream target suppressed by GPX4. These changes trigger irreversible plasma membrane rupture, exacerbating ferroptosis and contributing to male infertility [53]. In S8/*Gss*^−/−^ mice, decreased Gpx4 and increased Alox15 levels caused testicular ferroptosis, abnormal spermatogenesis, reduced sperm concentration, and morphological defects, ultimately leading to infertility [54]. Increasing the GSH level mitigated these effects by suppressing ROS production, inhibiting ferroptosis, and restoring spermatogenesis [55]. GPX4 reduces phospholipid hydroperoxides (PLOOHs) to their corresponding phospholipid alcohols (PLOHs), thereby inhibiting ferroptosis. GPX4 downregulation promotes germ cell ferroptosis, adversely affecting male fertility [49,56]. In NOA patients, *GPX4* downregulation is associated with ferroptosis [14]. Similarly, asthenozoospermic patients exhibit GPX4 and SLC7A11 downregulation, which correlates with increased iron deposition and impaired sperm motility [15]. Enhancing Gpx4 expression alleviates ferroptosis-related infertility [51]. The ferroptosis inhibitor Fer-1 significantly reduced lipid ROS and restored Gpx4 levels in GC-2spd cells [21]. Additionally, fecal microbiota transplantation upregulated Gpx4, inhibited ferroptosis, and restored spermatogenesis in mice [57].

Critical factors, including transferrin (TF) and Nrf2, mediate ferroptosis in testicular cell. Cells regulate iron intake through TF and transferrin receptor (TFRC). Experimental studies have shown that knockdown of *Tfrc* can block iron-dependent cell death by reducing mitochondrial and intracellular Fe^2+^ levels in TM4 cells, suggesting that iron overload plays a critical role in ferroptosis in these cells [23]. Additionally, Nrf2 is a key regulator of ferroptosis, directly or indirectly upregulating GPX4 expression and enhancing ferroptosis resistance. Nrf2 promotes intracellular cysteine enrichment via system Xc^−^, enhances GPX4 synthesis, and reduces its degradation [53]. Cancer cells with elevated NRF2 levels exhibit increased ferroptosis resistance [58]. NRF2 is essential for maintaining iron homeostasis and inhibiting ferroptosis through GPX4 upregulation [58]. The absence of Nrf2 downregulates GPX4, impairing the clearance of phospholipid hydroperoxides (PLOOHs) and causing excessive lipid peroxide accumulation, thereby triggering ferroptosis in spermatogenic cells [59]. Functional polymorphisms in *NRF2* promoters are strongly associated with defective spermatogenesis in humans [60,61]. And sperm from oligozoospermic men show lower NRF2 and GPX4 expression compared to healthy controls, further supporting this mechanism [62,63]. Ferroptosis represents a pivotal pathway in iron-induced male infertility, integrating the accumulation of lipid ROS and lipid peroxidation, mitochondrial dysfunction, and genetic dysregulation.

## 3. Cuproptosis

### 3.1. Cuproptosis in Male Infertility

Copper (Cu) is an essential trace element crucial for male reproductive health, particularly in spermatogenesis. Elevated Cu levels in seminal plasma have been associated with impaired sperm motility and abnormal morphology, likely due to its involvement in oxidative stress. Subfertile men exhibit significantly higher Cu concentrations in their serum, seminal plasma, and urine compared to fertile controls, which correlate with lower sperm count and motility [64,65]. Furthermore, Yin et al. [66] demonstrated that high Cu levels in seminal plasma were associated with increased oxidative stress markers, such as MDA, and a higher risk of idiopathic oligoasthenoteratozoospermia. The Cu/Zn ratio in seminal plasma also appears to be a crucial factor in male infertility. Akinloye et al. [67] noted that infertile men exhibit a lower Cu/Zn ratio in seminal plasma, suggesting that an imbalance between these trace elements could impair sperm function. Collectively, these studies highlight the importance of regulating Cu levels to maintain sperm health and fertility. Therefore, determination of Cu levels in serum, seminal plasma, and urine during infertility investigations is recommended. The toxicity of Cu is also related to the destruction of enzymes containing iron-sulfur (Fe-S) clusters. Multiple studies have shown that Cu blocks the formation of Fe-S clusters and disrupts the stability of Fe-S proteins. The accumulation of Cu ions can trigger oxidative stress. Excessive Cu leads to impaired sperm production and infertility through ROS generation and loss of mitochondrial membrane potential [68].

### 3.2. Possible Mechanisms of Cuproptosis in Male Infertility

Cuproptosis, a Cu-dependent form of programmed cell death, has emerged as a critical mechanism underlying male infertility, particularly under conditions of Cu overload. This process is characterized by mitochondrial dysfunction, oxidative stress, and aggregation of lipoylated proteins, leading to irreversible germ cell damage. Studies show that Cu nanoparticles (CuNPs) and copper sulfate (CuSO₄) induce oxidative stress, leading to increased MDA levels and apoptosis in testicular cells. These effects reduce sperm quality, disrupt hormone levels, and impair testicular function, with higher doses resulting in more severe damage [69,70,71]. CuNPs, for instance, downregulate steroid receptors, which inhibits germ cell proliferation and compromises spermatogenesis [70,72]. In a murine model of Cu overload (CuSO_4_ exposure), Zhang et al. [73] demonstrated that elevated Cu levels disrupted testicular Cu homeostasis, upregulated cuproptosis-related factors (e.g., Fdx1), and induced mitochondrial ATP depletion, resulting in apoptosis of spermatogenic cells. Similarly, Guo et al. [74] observed that Cu-induced oxidative stress triggered DNA damage and suppressed DNA repair pathways, further exacerbating germ cell apoptosis. Mitochondrial damage, evidenced by reduced membrane potential and increased ROS production, was also linked to autophagy activation in pig testes exposed to long-term Cu exposure [75]. These findings highlight the centrality of mitochondrial dysfunction in cuproptosis-mediated spermatogenic failure, with oxidative stress acting as a key amplifier of cellular damage.

Beyond direct cellular damage, Cu overload perturbs immune homeostasis and epigenetic regulation in the testicular microenvironment. Zhao et al. [76] identified upregulated expression of cuproptosis-related genes (e.g., NLR family pyrin domain-containing 3 (*NLRP3*), solute carrier family 31 member 1 (*SLC31A1*)) in human spermatogenic dysfunction, correlating with immune cell infiltration, particularly resting memory CD4+ T cells. This inflammatory response, driven by toll like receptor 4(TLR4) /NF-κB signaling, synergizes with PANoptosis (pyroptosis, apoptosis, and necroptosis) to exacerbate testicular damage [77]. Epigenetic modifications further compound these effects. Shaoyong et al. [78] reported that Cu oxide nanoparticles reduced H3K9me3 levels in spermatozoa, impairing mitochondrial energy metabolism and transgenerationally transmitting asthenospermia. Additionally, Huang et al. [79] linked Cu overload to *lncRNA:CR43306* deficiency in Drosophila, which disrupted spermatid differentiation via ferroptosis-cuproptosis crosstalk. These studies underscore the interplay between Cu toxicity, immune dysregulation, and epigenetic instability in male infertility, suggesting that therapeutic strategies must address both direct and systemic mechanisms.

Current research highlights potential interventions to mitigate Cu-induced spermatogenic damage. Chen et al. [80] demonstrated that GSH supplementation alleviated oxidative stress and apoptosis in Wilson’s disease models by restoring mitochondrial function. Similarly, retinoic acid rescued meiosis initiation in NSC319726-exposed mice, independent of cuproptosis, by compensating for retinol dehydrogenase deficiency [81]. Natural compounds like propolis and Tribulus terrestris extract attenuated Cu toxicity by enhancing antioxidant defenses and steroidogenesis [82,83]. Cu chelators (e.g., penicillamine, DMSA) and TLR4 inhibitors (e.g., eritoran) also showed efficacy in reducing testicular Cu deposition and inflammation [77,84,85]. However, challenges persist in managing chronic exposure and transgenerational effects. For instance, Nicy et al. [72] observed persistent testicular impairment in mice even after cessation of Cu nanoparticle treatment, indicating irreversible damage at high doses.

Cuproptosis represents a pivotal pathway in Cu-induced male infertility, integrating mitochondrial dysfunction, oxidative stress, immune activation, and epigenetic dysregulation. Copper homeostasis has been proven to regulate various cellular activities in cancer, and Cu chelators have huge potential for treatment [86]. Clinical trials on bis-choline tetrathiomolybdate (TTM), a copper chelator, demonstrate reduced copper uptake in Wilson’s disease and tolerability in breast cancer patients [87,88]. While therapeutic advances such as antioxidants and chelators show promise, gaps remain in understanding tissue-specific Cu transporters (e.g., high-affinity Cu transporter CTR1(CTR1), ATPase copper transporting alpha(ATP7A)) and their roles in germ cell survival [89,90,91]. Bioinformatic analyses revealed that elevated *FDX1* expression correlates with poor prognosis in glioma and gastric cancer, suggesting its role in copper-driven cell death [92,93]. However, the clinical relevance of cuproptosis-related biomarkers (e.g., *NRF2*) in human nonobstructive azoospermia warrants validation in larger cohorts [94]. Multidisciplinary approaches combining omics technologies, targeted drug delivery, and epigenetic interventions will be essential to translate these findings into effective therapies for Cu-related reproductive disorders.

## 4. Autophagy

### 4.1. Autophagy Exists in Clinical Diseases Related to Male Infertility

Autophagy is a conserved lysosome-dependent process wherein cytoplasmic components (e.g., damaged organelles) are sequestered within double-membraned autophagosomes and degraded to maintain cellular homeostasis. In autophagy, multiple key factors work together to ensure autophagosome formation, expansion, and fusion with lysosomes. Beclin-1(BECN1) initiates autophagosome formation by interacting with the phosphatidylinositol 3-kinase complex. Sequestome 1 (sequestome 1, also known as p62) recruits proteins to participate in the formation of autophagosomes.Autophagy-related 7 (ATG7), an E1-like activating enzyme, promotes autophagosome membrane expansion by facilitating autophagy-related 5 (ATG5) and ATG12 (autophagy-related protein 12) binding. ATG5 and ATG12 form a complex that binds with autophagy-related 16-like 1 (ATG16L1), further extending and closing the autophagosome membrane. Microtubule-associated protein 1 light chain 3 (LC3 proteins), markers of autophagosome membranes, convert from soluble LC3-I to membrane-bound LC3-II, integrating LC3-II into autophagosome membranes to aid their formation and expansion. Lysosome-associated membrane protein 2 (LAMP2), a crucial lysosomal membrane protein, is involved in the fusion of autophagosomes with lysosomes. GABA type A receptor-associated protein (GABARAP) is a ubiquitin-like protein belonging to the ATG8 family, involved in autophagosome formation, expansion, and fusion with lysosomes [95,96,97]. PTEN-induced putative kinase 1 (PINK1), a serine/threonine kinase, is a sensor of mitochondrial damage and mitochondrial autophagy (mitophagy). These factors collaborate in autophagy to maintain cell homeostasis.

Autophagy has been implicated in clinical diseases associated with male infertility. The *CT55* gene of two infertile siblings had a semizygous nonsense mutation, with symptoms of insufficient sperm personalization, excessive cytoplasmic residue, and defective acrosome development. *Ct55*-knockout male mice also showed infertility and sperm destruction. Endoplasmic reticulum (ER)- and ribosome-associated proteins were increased in the testes of patients and mice, while both LAMP2 and GABARAP were significantly reduced, indicating decreased autophagy activity. Further research showed that CT55 participates in autophagy through its interaction with LAMP2 and GABARAP, thereby affecting spermatogenesis [98]. In addition, patients with NOA have lower levels of *MIR-188-3p* and higher levels of *ATG7* in testicular tissue. A further study revealed that the inhibition of *MIR-188-3p* directly targeted *ATG7* to increase its expression and promote autophagy, impairing spermatogenesis [99]. In a study of infertility mediated by varicocele, the levels of autophagy markers ATG7 and LC3 proteins in the semen of patients were significantly higher than in the semen of fertile men, suggesting that autophagy is overactivated in male infertility caused by varicocele [100]. In the testicular tissue of rats with varicocele, the levels of autophagy-related protein LC3 and LC3-II/LC3-I ratio were significantly increased, further confirming this observation [101]. In conclusion, autophagy levels are abnormal in the clinical disease of male infertility mentioned above, which affects spermatogenesis.

Environmental pollutants disrupt spermatogenesis by dysregulating autophagy through oxidative stress, endoplasmic reticulum (ER) stress, and alterations in signaling pathways. Mycotoxins, such as zearalenone (ZEA) and aflatoxin B1 (AFB1), inhibit the PI3K/Akt/mTOR axis, activating autophagy in Sertoli and germ cells. ZEA induced oxidative stress in goat Sertoli cells, elevating the LC3-II/LC3-I ratio, reducing p62, and promoting lysosomal biogenesis via transcription factor EB (TFEB) nuclear translocation [102]. Similarly, AFB1 increased autophagosome formation and upregulated LC3, Beclin-1, and Atg5 in mouse testes by suppressing PI3K/Akt/mTOR, with oxidative stress further exacerbating autophagy [103]. Heavy metals like Cd and Cu exhibit dual roles: Cd hyperactivates autophagy by elevating LC3, Atg5, and Lamp2, disrupting the blood–testis barrier (BTB) through vimentin degradation [104,105], while Cu activates AMPK-mTOR-dependent autophagy to mitigate oxidative damage, paradoxically protecting sperm motility [106]. Nanoparticles and endocrine disruptors further modulate autophagy. Zinc oxide nanoparticles (ZnO NPs) induce ROS-mediated autophagy in GC-1 spermatogonia, increasing LC3-II and Atg5 [107]. Di-(2-ethylhexyl) phthalate (DEHP) upregulates PINK1/PARKIN and XBP1, driving mitophagy in spermatogonia via ER stress [108], whereas tributyltin chloride (TBTCL) inhibits autophagy, triggering apoptosis in Sertoli cells [109]. Perfluorooctanoic acid (PFOA) blocks autophagosome-lysosome fusion by downregulating α-SNAP, impairing BTB integrity [110]. Ethanol selectively activates autophagy in stage VII–VIII Sertoli cells, marked by LC3 accumulation and PINK1-mediated mitochondrial clearance, disrupting spermatogenic cycles [111]. These findings underscore a unifying mechanism: pollutants induce oxidative/ER stress to dysregulate autophagy, either promoting excessive degradation (e.g., Cd, ZEA) or impairing flux (e.g., PFOA), ultimately compromising germ cell viability and BTB function. Targeting autophagy regulators such as mTOR or TFEB may offer therapeutic strategies to counteract pollutant-induced male infertility.

Autophagy plays a crucial role in treating male infertility and spermatogenic disorders induced by lifestyle factors (e.g., obesity), drugs, or environmental factors. For example, Qiangjing tablets activated mitophagy in a mouse model of asthenozoospermia, improving sperm motility by enhancing the LKB1/AMPK/ULK1 signaling pathway [112]. Similarly, eprosartan activated autophagy in testicular torsion models by modulating the Beclin-1 and AMPK/mTOR pathways, mitigating testicular damage [113]. Clinical study have found that infertile oligoasthenoteratozoospermic men associated with varicocele have significantly higher cystatin C levels in semen than healthy males. And six months after surgery, the seminal cystatin C levels were significantly lower than pre-operative levels.This implied that cystatin C may ameliorate male infertility due to varicocele by modulating autophagy [114]. Sancai Lianmei granules improve sperm density and viability in obese mice by inhibiting excessive autophagy [115]. In drug-induced infertility, such as busulfan or paclitaxel exposure, autophagic activity is dysregulated, and supplementation with fatty acids like octanoic acid or compounds like naringin can reverse these effects, improving spermatogenesis and testicular function [116,117]. Moreover, thymoquinone alleviates cisplatin-induced testicular damage by modulating autophagic and apoptotic pathways, restoring sperm quality [118]. Environmental pollutants like bisphenol A and Cd also disrupt spermatogenesis via autophagy dysregulation. Lactoferrin and MSC-derived exosomes help to restore autophagic flux, reducing oxidative stress and improving testicular function [119,120]. Berberine nanostructured biloalbuminosomes and M1 nanostructured biloalbuminosomes ameliorated Cd- and Pb-induced testicular and prostate damage in rats by attenuating oxidative stress-induced excessive autophagy [121]. Ginkgo biloba extract attenuated insecticide deltamethrin-induced testicular damage in mice by upregulating SKP2, a key regulator of Beclin-1-independent autophagy, and inhibited Beclin-1 activation to regulate autophagy [122]. These studies highlight the potential of autophagy modulation as a therapeutic strategy for male infertility caused by various factors.

### 4.2. Mechanism of Autophagy in Male Infertility

Autophagy plays a pivotal role in meiosis and early spermatogenesis, ensuring genomic integrity and germ cell survival. *Forkhead box J2 (Foxj2)*, a transcription factor critical for meiotic progression, regulates DNA double-strand break (DSB) repair and chromosomal synapsis. Germline deletion of *Foxj2* in mice resulted in meiotic arrest due to defective DSB repair and reduced expression of repair factors (e.g., Rad51, Brca1), highlighting its role in maintaining meiotic fidelity [123]. Paradoxically, *Foxj2* overexpression activated aberrant autophagy by upregulating Lamp2A, a chaperone-mediated autophagy receptor, leading to excessive lysosomal degradation and meiotic failure [124]. Concurrently, adenine nucleotide translocase 4 *(A**nt**4)*, essential for mitochondrial ADP/ATP transport, is indispensable for spermatocyte survival. *Ant4*-deficient mice exhibited spermatocyte apoptosis, mitochondrial dysfunction, and disrupted autophagic flux via the AKT-AMPK-mTOR pathway, linking energy metabolism to autophagy regulation [125,126]. These findings underscore the delicate balance of autophagy in meiosis: while basal autophagy supports DSB repair and mitochondrial homeostasis, dysregulation—either deficiency or hyperactivation—compromises germ cell viability.

During spermiogenesis, autophagy facilitates cytoplasmic remodeling and organelle clearance, processes critical for sperm individualization. ATG5, a core autophagy protein, mediates autophagosome formation and sperm elongation. Conditional knockout of *Atg5* in germ cells disrupted spermatid development, resulting in residual cytoplasmic bodies, malformed acrosomes, and infertility [127]. Similarly, BECN1, a key regulator of autophagosome nucleation, is essential for spermiation and mitochondrial organization. Germline *Becn1* deletion impaired autophagy, leading to abnormal sperm motility, acrosome defects, and epididymal epithelial loss [128]. Beyond canonical autophagy, ***A**tg**5*** also participates in non-autophagic processes such as lysosomal repair via ESCRT machinery, which is critical for maintaining sperm structural integrity [129]. Additionally, cathepsin B (CTSB), a lysosomal protease, modulates autophagy–apoptosis crosstalk during sperm maturation. *Ctsb*-null mice exhibited inhibited autophagy, increased apoptosis, and epididymal epithelial damage, emphasizing CTSB’s role in balancing degradation and survival pathways [130,131]. mTOR acts as a core factor regulating spermatogenesis, and the use of mTOR inhibitors such as rapamycin can improve spermatogenesis by activating autophagy in the presence of autophagy deficiency [132,133]. These studies collectively demonstrate that autophagy coordinates organelle clearance, energy homeostasis, and stress responses to ensure functional sperm production.

Autophagy in spermatogenesis is tightly regulated by phosphorylation events and metabolic signaling. BECN1 phosphorylation fine-tunes autophagosome assembly, integrating stress signals to maintain pro-survival autophagy [134]. For instance, oxidative stress and ER stress (e.g., induced by environmental toxins) modulate BECN1 activity, impacting spermatogonial survival and meiotic progression. However, unresolved questions persist: (1) How do *Foxj2* and *Ant4* mechanistically intersect with autophagy pathways? (2) What role do non-canonical autophagy proteins, such as ATG5 in lysosomal repair, play in sperm quality? (3) Can CTSB or LAMP2A be targeted and lysosomal stabilizers rescue autophagy dysregulation in infertility by mitigating environmental or genetic insults? Understanding these mechanisms will advance strategies to address male infertility linked to autophagy dysfunction.

## 5. Pyroptosis

### 5.1. Pathological Roles of Pyroptosis in Diverse Etiologies of Male Infertility

Pyroptosis, a pro-inflammatory programmed cell death driven by gasdermin family proteins (e.g., gasdermin D (GSDMD)), is characterized by plasma membrane pore formation, cytokine release (e.g., interleukin-1β(IL-1β)), and osmotic lysis. It plays dual roles in testicular immune defense and pathological inflammation. Clinical and preclinical studies implicate pyroptosis as a hallmark of spermatogenic failure. In human spermatogenic dysfunction, pro-pyroptotic factor *CASPASE-4*(*CASP4*) and anti-pyroptotic factor *GPX4* are identified as key regulators, with *CASP4* negatively correlating with Johnsen scores and *GPX4* exerting protective effects [135]. Varicocele-associated infertility involves oxidative mtDNA activating the cGAS/STING pathway, which promotes NLR family pyrin domain-containing 3 inflammasome (NLRP3 inflammasome) activity and pyroptosis in Sertoli cells [136]. Seminal inflammasome markers (e.g., IL-1β, cysteine-aspartic acid protease-1(caspase-1)) are elevated in varicocele patients, and varicocelectomy reduces these mediators, thereby improving semen quality [137]. Obesity and metabolic disorders exacerbate pyroptosis through NLRP3 inflammasome activation. A high-fat diet (HFD) elevated testicular high mobility group box 1 (HMGB1) and NLRP3 levels, impairing sperm parameters, whereas zinc supplementation mitigated oxidative stress and inflammation [138]. *Blautia wexlerae* depletion in obesity disrupts acetate-mediated suppression of NLRP3, whereas its restoration attenuates pyroptosis and improves fertility [139]. These data underscore pyroptosis as a pathological nexus in male infertility, driven by oxidative stress, cytokine dysregulation, and environmental insults.

The environmental drivers of testicular pyroptosis are various. Environmental toxins such as fluoride and 1,2-dichloroethane (1,2-DCE) induce pyroptosis via IL-17A signaling and piRNA-mmu-1019957/IRF7 pathways, respectively, thereby impairing germ cell viability [140,141]. Co-exposure to microcystin and nitrite synergistically enhances mitochondrial ROS (mtROS)-dependent pyroptosis and apoptosis, exacerbating spermatogenic disorders [142]. Silver nanoparticles (AgNPs) induce NF-κB/NLRP3-driven inflammation in spermatogonia, highlighting their reprotoxic potential [143]. Additionally, lead exposure triggers pyroptosis-mediated fibrosis via NF-κB and caspase-1 activation, further compromising testicular function [144]. These studies emphasize the interplay between extrinsic factors and intrinsic inflammasome pathways in driving male reproductive dysfunction. The different forms of cell death implicated in environmental pollutant-induced male infertility are summarized in Table 1.

### 5.2. Molecular Mechanisms of Pyroptosis and Its Role in Testicular Homeostasis

Under physiological conditions, pyroptosis aids in eliminating infected or damaged cells, maintaining immune homeostasis [147]. However, pathological activation of the NLRP3 inflammasome, triggered by oxidative stress or mitochondrial dysfunction, disrupts spermatogenesis by cleaving caspase-1/4/11, which processes gasdermin D (GSDMD) to form membrane pores, releasing pro-inflammatory cytokines (IL-1β, IL-18) and cellular debris [148,149]. For instance, in Sertoli cell-only syndrome (SCOS), upregulated *CASP1* and *CASP4* in Leydig and Sertoli cells drive pyroptosis, which correlates with germ cell loss and elevated inflammatory markers [150]. Similarly, Cd exposure induces absent in melanoma 2 inflammasome (AIM2)-dependent pyroptosis via m6A-modified *Lonp1*, which disrupts mitochondrial proteostasis and releases mitochondrial DNA (mtDNA) into the cytoplasm, exacerbating testicular injury [145]. These findings highlight the NLRP3/GSDMD axis as central to pyroptosis-mediated testicular damage, with caspase activation and mitochondrial dysfunction as critical contributors. Pyroptosis emerges as a pivotal mechanism linking inflammation, oxidative stress, and environmental toxins to male infertility. Moreover, context-dependent regulators like miRNAs (e.g., *MiR-153*, *MiR-195-5p*) and mitochondrial dynamics modulate outcomes [151,152].

Targeting pyroptosis pathways offers promising interventions. Melatonin and Bifidobacterium suppress NLRP3 and caspase-1/11, alleviating fluoride- and 1,2-DCE-induced damage [140,141]. Chlorogenic acid inhibits cGAS/STING-mediated NLRP3 activation in varicocele, restoring mitochondrial homeostasis [136]. Pharmacological GSDMD inhibitors (e.g., disulfiram) and gut microbiota modulators like Guijiajiao improve spermatogenesis by reducing inflammation [153,154] (Table 2). Monosodium glutamate upregulates pyroptotic markers (NLRP3, caspase-3), which resveratrol reverses via antioxidant and anti-pyroptotic effects [146]. NLRP3 inhibitors (e.g., MCC950) attenuate COVID-19-associated testicular inflammation [155]. Virtual screening identified FDA-approved drugs (ChEMBL IDs: 4297185, 1201749, 1200545) targeting NLRP3 for varicocele treatment [156]. Therapeutic strategies targeting pyroptosis components show preclinical efficacy, yet clinical validation is imperative. Future research should prioritize cell-specific delivery systems for pyroptosis inhibitors and combinatorial therapies targeting oxidative stress and mitochondrial integrity.

However, challenges remain: the tissue-specific roles of gasdermin isoforms (e.g., gasdermin E ) in testicular pyroptosis are underexplored, and crosstalk between pyroptosis, apoptosis, and autophagy in germ cells requires further study [157,158]. Understanding pyroptosis’s dual roles—pathological vs. protective—in testicular physiology will refine interventions, ultimately improving fertility outcomes for affected individuals.

## 6. The Relationship Between Cell Death Forms in Male Infertility

### 6.1. The Dual Role of SLC7A11 in Ferroptosis and Disulfidptosis

Disulfidptosis is a novel form of cell death. In recent years, with the in-depth study of cell death mechanisms, the role of disulfidptosis in male infertility has gradually been revealed. High expression of SLC7A11 (responsible for cysteine uptake) promotes the depletion of nicotinamide adenine dinucleotide phosphate (NADPH). Under glucose starvation, SLC7A11-mediated cysteine uptake further depletes intracellular NADPH, leading to its depletion and the accumulation of disulfide bonds. This results in the formation of disulfide bonds between actin cytoskeleton proteins and the breakdown of the actin filament (F-actin) network [159,160], which leads to disulfidptosis.

SLC7A11, a critical component of the cystine/glutamate antiporter system Xc^−^, plays a paradoxical role in regulating ferroptosis and disulfidptosis, two distinct forms of regulated cell death implicated in male infertility. Ferroptosis, driven by iron-dependent lipid peroxidation, is tightly linked to SLC7A11’s function in maintaining redox homeostasis. By importing cystine for GSH synthesis, SLC7A11 supports GPX4 activity to neutralize lipid peroxides, thereby protecting testicular cells from oxidative damage [33,161,162,163]. For instance, Cr (VI) and DEHP exposures downregulate SLC7A11, deplete GSH, and induce ferroptosis in Sertoli and Leydig cells, leading to seminiferous tubule atrophy and impaired spermatogenesis [161,162,164]. Conversely, disulfidptosis arises from excessive cystine accumulation under glucose deprivation, where SLC7A11 overexpression overwhelms NADPH-dependent reduction capacity, causing toxic disulfide stress and membrane rupture [165,166]. In Bisphenol F(BPF)-induced testicular injury, obesity-associated protein(FTO) deficiency disrupts m6A-mediated SLC7A11 mRNA stability, exacerbating disulfidptosis via YTH domain family 2(YTHDF2)-dependent decay [165]. These findings highlight SLC7A11’s dual regulatory role: its downregulation promotes ferroptosis by limiting antioxidant defenses, while its overexpression under metabolic stress triggers disulfidptosis through cysteine overload.

### 6.2. The Facilitating Role of ROS in Ferroptosis and Cuproptosis

Common mechanisms in both ferroptosis and cuproptosis are oxidative stress and mitochondrial dysfunction. Both ferroptosis and cuproptosis trigger cell damage through ROS accumulation. In ferroptosis, Fe^2+^ generates hydroxyl radicals (OH•) through the Fenton reaction, resulting in lipid peroxidation [49,52]. In cuproptosis, excess Cu^2+^ directly produces ROS that disrupt mitochondrial membrane potential [68,73]. GSH depletion and downregulation of GPX4 in ferroptosis [15,54] and GSH supplementation in cuproptosis [80] suggest that both are dependent on the integrity of the antioxidant system. Ferroptosis leads to mitochondrial lipid peroxidation through GPX4 inhibition failure [53], whereas cuproptosis induces mitochondrial ATP depletion and loss of membrane potential via Cu ions [73]. Both involve the activation of mitochondrial permeability transition pores (mPTP), which triggers cell death [50,75].

Through cross-regulation at the molecular level, Nrf2 inhibits lipid peroxidation by upregulating GPX4 and SLC7A11 in ferroptosis [53,58], and in cuproptosis, activation of Nrf2 alleviates Cu-induced oxidative stress [80]. Whereas FDX1 regulates lipid peroxidation in ferroptosis [22] and is a key factor in Cu-dependent death in cuproptosis [73]. Translating these findings to male infertility, we hypothesize that FDX1 overexpression in testicular cells may exacerbate copper toxicity, impairing spermatogenesis. Future studies should explore urinary/serum copper levels and FDX1 activity in infertile males, potentially establishing FDX1 as a diagnostic biomarker or therapeutic target. Cu overload triggers cuproptosis through the mitochondrial TCA cycle, facilitating intracellular interactions with ferroptosis, thereby governing testicular aging [79]. Ferroptosis and cuproptosis cross-function in male infertility through oxidative stress, mitochondrial damage, and Nrf2 pathway dysregulation. In the future, it is necessary to further explore the interaction mechanism of FDX1 and GPX4 molecules in the two types of death to develop precise intervention methods.

### 6.3. The Balance of Pyroptosis and Autophagy in Male Fertility

The balance between pyroptosis and autophagy is essential for male fertility. Autophagy inhibits pyroptosis hyperactivation by degrading the NLRP3 inflammasome, e.g., chlorogenic acid inhibits NLRP3 through the cGAS/STING pathway and ameliorates varicocele [136]. Conversely, IL-1β released through pyroptosis may inhibit autophagy, creating a vicious cycle [137]. Pyroptosis and autophagy share the upstream signaling pathways mTOR and ROS. mTOR inhibits the activation of autophagy (e.g., the AMPK/mTOR axis regulates autophagy) while inhibiting NLRP3 inflammasome activity [104,106]. ROS activates the NLRP3 inflammasome to trigger pyroptosis (e.g., mtROS accumulation in varicocele [136]), while inducing autophagy to clear the damaged component. If autophagy is insufficient (e.g., antioxidant deficiency), pyroptosis predominates and spermatogenic cell death (e.g., microcystins synergize with nitrite) [142]. Crossover mechanisms occur at the molecular level. Cleavage fragments of GSDMD may interfere with autophagosome maturation by disrupting lysosomal membranes [149]. Synergy and conflict of blood–testis barrier function are induced by pyroptosis and autophagy. Sertoli cell pyroptosis directly disrupts the blood–testis barrier, leading to immune cell infiltration (e.g., lead exposure activates pyroptosis via NF-κB/caspase-1) [144]. Autophagy maintains cellular junctions by degrading aberrant proteins such as connexin 43. For example, perfluorooctanoic acid (PFOA) blocks autophagosome–lysosome fusion, impairing the blood–testis barrier [110].

## 7. Conclusions

This review delineates the pathogenic roles of ferroptosis, cuproptosis, autophagy, and pyroptosis in male infertility, highlighting their molecular cross-talk and environmental triggers. Key mechanisms include oxidative stress, mitochondrial dysfunction, and inflammasome activation, which disrupt spermatogenesis and BTB integrity (Figure 3). Therapeutic strategies targeting antioxidant defenses (e.g., Nrf2 activators), metal homeostasis, and inflammatory pathways demonstrate efficacy in mitigating testicular damage.

Future research must prioritize: (1) validating biomarkers (e.g., GPX4, FDX1) in clinical cohorts for early diagnosis; (2) developing precision therapies, such as nanoparticle-based delivery of pathway-specific inhibitors (e.g., GSDMD or FDX1-targeted drugs) or CRISPR editing to correct autophagy-related mutations; (3) elucidating tissue-specific regulators (e.g., gasdermin E in pyroptosis) and their interactions; (4) addressing the transgenerational effects of environmental toxins through epigenetic studies (e.g., H3K9me3 loss in CuONP exposure); (5) multi-omics integration to map the multi-cell death pathway dynamics across spermatogenic stages. Integrating multi-omics and AI-driven models will advance mechanistic understanding and therapeutic innovation, potentially improving fertility outcomes in affected populations.

## Figures and Tables

**Figure 1 ijms-26-03496-f001:**
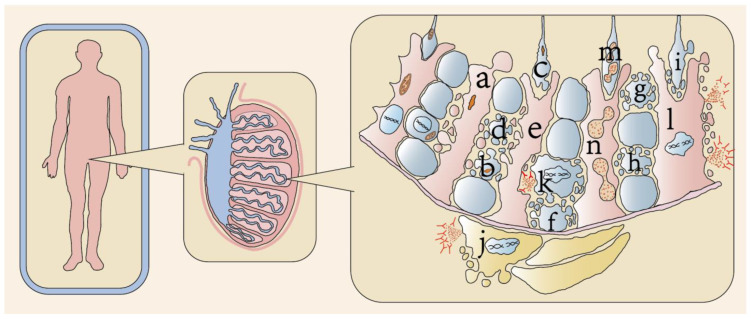
**Modes of cell death in the human testis.** The schematic illustrates distinct forms of cell death within testicular compartments. Color coding: blue (seminiferous tubules, housing germ cells), red (Sertoli cells), and yellow (Leydig cells). Specific cell death modalities are annotated as follows: a. ferroptosis of human Sertoli cell, b. ferroptosis of human spermatogonial cell, c. ferroptosis of human sperm cell, d. apoptosis of human spermatocyte, e. apoptosis of human Sertoli cell, f. apoptosis of human spermatogonial stem cell, g. apoptosis of human sperm cell, h. apoptosis of human spermatogonial cell, i. apoptosis of human sperm cell, j. pyroptosis of human Leydig cell, k. pyrosis of human spermatogonial cell, l. pyroptosis of human Sertoli cell, m. autophagy of human sperm cell, and n. autophagy of human Sertoli cell. The diagram was drawn using Adobe Illustrator 2023.

**Figure 2 ijms-26-03496-f002:**
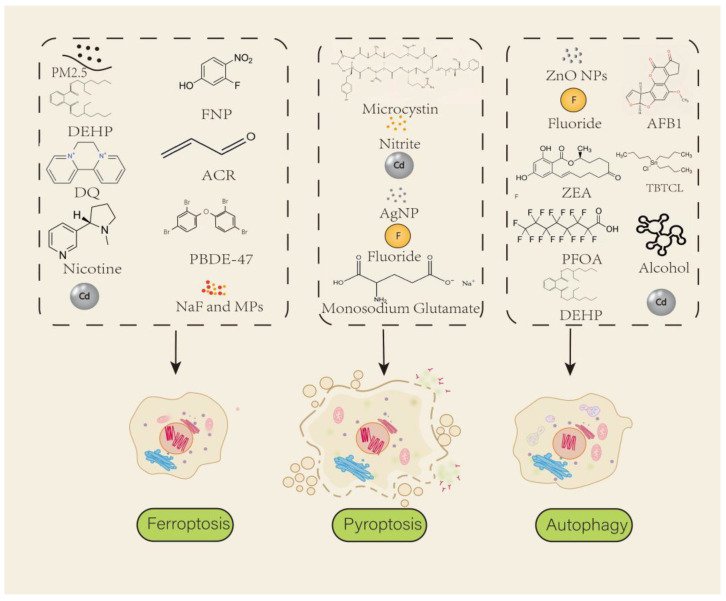
**Environmental factors cause cell death, inducing male infertility.** The diagram illustrates three types of cell death caused by different environmental factors: ferroptosis, pyroptosis, and autophagy. The black arrows indicate that the substances within each box cause a type of cell death. PM2.5 (particulate matter 2.5); DEHP (plasticizer di-(2-ethylhexyl) phthalate); FNP (insecticide 3-fluoro-4-nitrophenol); DQ (herbicide 1,1′-ethyl-2,2′ -bipyridine dibromide); ACR (acrolein); PBDE-47(2,2′,4,4′-tetrabromodiphenyl ether); Cd (cadmium); NaF and MPs (the combined effects of sodium fluoride and microplastics); AgNP (silver nanoparticles); ZnO NPs (zinc oxide nanoparticles); AFB1 (aflatoxin B1); ZEA (zealenone); TBTCL (tributyl tin chloride); PFOA (perfluorooctanoic acid). The diagram was drawn using Adobe Illustrator 2023.

**Figure 3 ijms-26-03496-f003:**
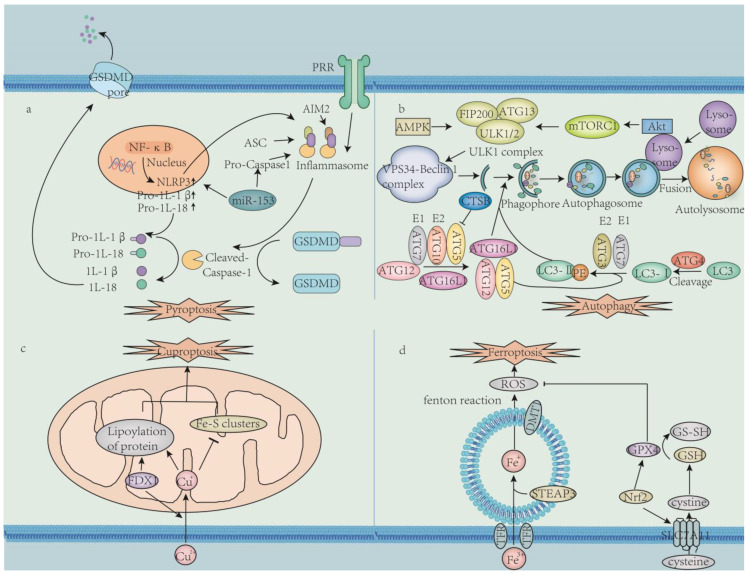
**Four cell death mechanisms in male infertility.** (**a**). Pyroptosis. Some signaling molecules or stimuli activate pattern recognition receptors, thereby activating the inflammasome, activating caspase-1, cleaving the GSDMD protein, perforating the cell membrane, causing cell death, and releasing inflammatory factors to amplify the inflammatory response. Black arrows indicate activation or upregulation; adaptor protein (ASC); pattern recognition receptors (NLRP3, AIM2); gasdermin D (GSDMD); cysteine-aspartic acid protease-1 (caspase-1); microRNA-153 (miR-153); nuclear factor kappa-light-chain-enhancer of activated B cells (NF-κB); inflammatory factors (IL-1β, IL-18). (**b**). Autophagy. After sensing external emergency signals, the cell activates AKT and AMPK or inhibits mTORC1, thereby activating the ULK1 complex and recruiting other autophagy-related proteins (ATG proteins) to initiate the formation of autophagosomes. The ATG5-ATG12-ATG16L1 complex is involved in membrane expansion, LC3-II is involved in autophagosome labeling, and mature autophagosomes fuse with lysosomes to form autolysosomes, degrading organelles and proteins. Excessive degradation or degradation disorders can lead to cell death. Autophagy-related genes (ATG); vacuolar protein sorting 34 (VPS34); Unc-51-like kinase 1 (ULK1); cathepsin B (CTSB); phosphatidylethanolamine (PE). (**c**). Cuproptosis. Cu+ inhibits Fe-S cluster protein synthesis and promotes protein lipid acylation. FDX1, ferredoxin 1. (**d**). Ferroptosis. Fe2+ produces ROS and thus causes ferroptosis. GPX4 inhibits ROS production and thus inhibits ferroptosis. TFR, transferrin receptor; STEAP3, six-transmembrane epithelial antigen of the prostate 3; DMT1, divalent metal transporter 1; ROS, reactive oxygen species; SLC7A11, solute carrier family 7 member 11; GSH, glutathione; GS-SH, oxidized glutathione disulfide; Nrf2, nuclear factor erythroid 2-related factor 2; GPX4, glutathione peroxidase 4. The diagram was drawn using Adobe Illustrator 2023.

**Table 1 ijms-26-03496-t001:** Different forms of cell death involved in male infertility caused by environmental pollutants.

Environmental Pollutant	Experimental Animal (Male)	Experimental Cell Type	Impact on Male Fertility	Cell Death-Related Phenomena	Form of Cell Death	Ref
Cadmium (Cd)	Institute of Cancer Research (ICR) mice	GC-1 spermatogonial cell line (GC-1)	Cd exposure during puberty damaged testicular development and spermatogenesis in mice in adulthood.	Cd caused iron overload, triggered ferroptosis in spermatogonia, and disturbed iron metabolism in spermatogonia.	Ferroptosis	[29]
	C57BL/6 mice aged 6 weeks	/	Disruption of seminiferous tubules, extensive degenerative vacuolization, and intercellular gaps between germ cells.	Elevated levels of LC3, p62, Atg7, Beclin-1, Atg5, and lysosomal membrane protein Lamp2.	Autophagy	[104]
	Sprague-Dawley (SD) rats aged 10 weeks	TM4 Sertoli cell line (TM4)	BTB disruption with irregular and open Sertoli cell junctions.	Accumulation of autophagic vacuoles and increased LC3-II levels.	Autophagy	[105]
	CD-1 mice	GC-1	Enhanced testicular hyperemia and vacuolation.	Elevated levels of N-GSDMD, cleaved caspase-1, and mature IL-1β	Pyroptosis	[145]
Copper (Cu)	ICR mice aged 8 weeks	GC-1	Reduction of spermatogonia and spermatocytes with vacuolar degeneration and necrosis of spermatogenic cells.	Upregulation of Beclin-1, Atg5-Atg12, Atg7, Atg3, and Atg16L, and an increase in the LC3-II/LC3-I ratio.	Autophagy	[106]
Plasticizer di (2-ethylhexyl) phthalate (DEHP)	ICR mice	TM4	Testicular atrophy decreased the organ coefficient of mouse testes and disrupted the BTB in Sertoli cells.	DEHP induces ferroptosis in Sertoli cells, and its metabolite MEHP induces functional injury, disrupts glutathione metabolism disorder in TM4 cells, and accelerates ROS generation and lipid peroxidation.	Ferroptosis	[23]
Acrolein (ACR)	/	TM4	/	Increased Fe^2+^ levels, lipid peroxidation in Sertoli cells.	Ferroptosis	[33]
2,2′,4,4′-Tetrabromodiphenyl ether (PBDE-47)	SD rats	TM4	Reduced sperm count, increased the percentage of abnormal sperm, disrupted BTB integrity, and impaired testicular development.	Upregulation of Fe^2+^ and MDA levels, and downregulation of GSH levels and Gpx4 and Slc7a11 protein levels in Sertoli cells and testes.	Ferroptosis	[27]
Cu sulfate (CuSO4)	ICR mice	/	Dose-dependent testicular pathological disturbances, with reduced Sertoli cell and spermatogenic cell numbers.	Elevated blood levels of Cu, Fdx1, and Slc7a11.	Cuproptosis	[73]
Cu oxide nanoparticles (CuONP)	BALB/c mice	/	Testis development disorder and damaged sperm function and capacitation.	Increased ROS and MDA levels, decreased GSH content.	Cuproptosis	[78]
Di-(2-ethylhexyl) phthalate (DEHP)	SD rats	GC-1 and GC-2 spermatocyte cell line (GC-2)	The testicular organ coefficient is low, the seminiferous epithelium is disordered, the number of germ cells is reduced, and the supporting cells are deformed.	Upregulated PINK1 and PARKIN levels, and LC3 and COX IV co-localization.	Autophagy	[108]
Zinc oxide nanoparticles (ZnO NPs)	/	GC-1	Decreased vitality of GC-1 spg cells in mice.	The protein content of LC3-II, the ratio of LC3-II/LC3-I, and the protein levels of ATG5 and Beclin-1 increased.	Autophagy	[107]
Tributyltin chloride (TBTCL)	/	Mouse Leydig cell line	Reduced cell vitality.	Decreased LC3-II and Beclin-1 levels, increased autophagic substrate p62 level.	Autophagy	[109]
Perfluorooctanoic acid (PFOA)	BABL/c mice aged 8 weeks	TM4	Disruption of BTB integrity, reduction in sperm motility and count, seminiferous tubule damage, and significant decrease in seminiferous epithelium height.	Upregulation of autophagy-related proteins LC3B and p62 in testicular Sertoli cells, and elevated levels of LC3-II/I, Beclin-1, and p62 in TM4 cells.	Autophagy	[110]
Zearalenone (ZEA)	/	Dairy goat Sertoli cells	Induction of oxidative stress and impairment of spermatogenesis.	Increased the LC3-II/I ratio and decreased the level of p62 protein.	Autophagy	[102]
Aflatoxin B1 (AFB1)	Kunming mice	/	Atrophy of seminiferous tubules, vacuolar changes in seminiferous epithelium, reduction in testicular interstitial cell and sperm count, significant increase in sperm deformity rate, significant decrease in sperm motility, and decrease in serum testosterone levels.	Increased expression of LC3, Beclin-1, Atg5, Atg12, Atg13, and p62.	Autophagy	[103]
Cigarette/cigarette smoke condensate (CSC)	Semen samples	GC-2	Higher abnormalities of sperm viability and sperm progressive motility.	Heavy smokers: Decreased GSH levels, increased lipid ROS and iron levels. CSC treatment effects: Decreased GSH level and GPX4 protein level, increased lipid ROS and iron levels in GC-2spd cells.	Ferroptosis	[21]
Monosodium glutamate	Wistar rats	/	Testicular degeneration, decreased testosterone, FSH, and LH levels, and abnormal sperm morphology.	Elevated the levels of GSK-3β, NLRP3, caspase-1, and IL-1 β.	Pyroptosis	[146]
Ethanol	Adult Wistar rats	/	Inhibition of androgen receptor in testicular somatic cells.	Elevated expression of LC3, accumulation of PINK1 and large lipid droplets.	Autophagy	[111]
PM2.5	SD rats	/	Decreased the number of sperm in the testes, induced mitochondrial damage in testicular cells during sexual maturity in juvenile male rats, and disrupted iron metabolism.	Decreased Gpx4 expression, reduced Slc7a11 expression.	Ferroptosis	[25]
	/	TM-3 cells, mouse Leydig cell line	Dose-dependent decrease in TM-3 cell viability.	Significant lipid peroxidation, elevated MDA levels, and increased ferrous content.	Ferroptosis	[22]
Fluoride (NaF) microplastics (MPs)	BALB/c mice	TM4	Testicular injury, decreased Sertoli cell number, and functional impairment in mice.	Gpx4 and downregulation in PS-MP and NaF + PS-MP groups.NaF and PS-MP co-exposure induces ferroptosis in Sertoli cells.	Ferroptosis	[28]
Microcystin and nitrite	Balb/c mice	GC-1 and TM4	Decreased gonadal index and testosterone levels, reduced sperm density and survival rate, lowered number of spermatogonia.	Increased caspase-1, N-GSDMD, and NLRP3 levels.	Pyroptosis	[142]
Silver nanoparticles (AgNPs)	/	GC-1 and GC-2	/	Increased NF–κB and NLRP3 protein levels.	Pyroptosis	[143]
Fluoride	C57BL/6J mice aged 6 weeks	/	Decreased sperm number and vitality, increased sperm abnormalities, reduced spermatogenic cell count, loosened testicular structure, and widened seminiferous tubule spacing.	Upregulation of IL-18, IL-1β, LDH, NLRP3, AIM2, PYRIN, NLRP1, ASC, GSDMD, IL-1β, IL-18, Caspase-11, and Caspase-3.	Pyroptosis	[140]
1,2-Dichloroethane	Mice	GC-2	Testicular apoptosis, sperm malformation, and decreased sperm concentration.	Elevated levels of NLRP3, GSDMD, N-GSDMD, Caspase-1, IL-1β, ASC, and IL-18, and formation of membrane pores in GC-2 spd cells.	Pyroptosis	[141]

**Table 2 ijms-26-03496-t002:** The potential mechanisms of key factors involved in various forms of cell death in male infertility.

Experimental Animal	Experimental Cell Type	Form of Cell Death	Phenomena	Possible Mechanism	Refs
/	*Fdx1* knockdown model in TM-3 cells	Ferroptosis	The reduction in cell viability was alleviated, lipid peroxidation and cellular iron significantly declined after FDX1 knockdown.	Upregulation of Fdx1 can promote ROS accumulation and ferrous overload in Leydig cells, induce lipid peroxidation, and lead to ferroptosis. After knocking down *Fdx1*, lipid peroxidation and cellular iron were significantly reduced, improving PM_2.5_-induced lipid peroxidation and ferrous accumulation.	[22]
*Nrf2^−/−^* male mice	/	Ferroptosis	Decrease in fertility, decreases in sperm concentration and motility, inhibition of ferroptosis attenuated spermatogenic cell damage in the testes and epididymis, and increases in sperm concentration and motility.	Gpx4 and Slc7a11 significantly decreased.	[59]
S8/*Gss^−^*^/^*^−^* male mice	/	Ferroptosis	There was a decline in Gpx4 and an increase in Alox15 levels observed in 8-month-old S8/Gss^−/−^ mice, resulting in the accumulation of ROS, lipid peroxidation, and ultimately testicular ferroptosis.	Gss deficiency caused ferroptosis in the testes of mice, testicular ferroptosis may cause meiosis disruption and acrosome heterotopia. The resulting aberrant sperm showed lower concentrations and abnormal morphology, leading to reduced fertility.	[54]
*Atg5^flox/flox^*; *Stra8-iCre*	/	Autophagy	70% of mice are infertile, with reduced sperm count and motility and impaired sperm morphology.	The absence of ATG5 significantly reduces the expression of testicular LC3A/B-II, increases the expression of autophagy receptor SQSTM1/p62, and reduces testicular autophagy activity.	[127]
*Becn1^flox/Δ^*; *Stra8-Cre*	/	Autophagy	The fertility of mice decreased, with a significant decrease in sperm motility and count, as well as abnormalities in sperm morphology and structure.	The absence of Beclin-1 leads to the accumulation of p62, a decrease in Atg5, and inhibition of the autophagic process.	[128]
*Gsdmd*^fl/fl^*Cx3cr1-*cre	/	Pyroptosis	Improved sperm quality and reduced the expression of pro-inflammatory cytokines IL-1 β and TNF-α.	Specific knockout of GSDMD in macrophages can reduce the expression of pro-inflammatory cytokines IL-1 β and TNF-α and improve testicular injury induced by orchitis.	[153]

## Data Availability

The data are available upon request from the corresponding author.

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
