# Peer review of "Interplay of Ferroptosis, Cuproptosis, Autophagy and Pyroptosis in Male Infertility: Molecular Crossroads and Therapeutic Opportunities"

_ijms, 2025, doi:10.3390/ijms26083496_

Round 1

Reviewer 1 Report

Comments and Suggestions for Authors

This article provides a thorough examination of the cell death mechanisms — ferroptosis, cuproptosis, autophagy and pyroptosis — associated with male infertility.

The section on therapeutic interventions is comprehensive but needs to be more detailed, particularly in terms of analysing preclinical and clinical studies. In particular, a more detailed examination of ferroptosis inhibitors, NLRP3 antagonists and other treatments would strengthen the discussion. The inclusion of data from clinical trials or case studies would further validate the proposed pathways and increase their credibility.

In addition, the clarity of the illustrations, especially Figure 1, should be improved. The current layout makes it difficult to distinguish the individual components. Increasing the font and improving the general clarity would make it easier to interpret the illustrations. In addition, the section "Disulfide death" in Figure 3 should be removed as it is not mentioned in the text.

  • Line 64 – systematically instead systemativally
  • Line 124 - it would be useful to add here that Ferrostatin-1 is a ferroptosis inhibitor.
  • Line 170 – transferrin abbreviation here, not in line 171
  • Line 474 – cuproptosis vs cuproposis

Author Response

Response to Reviewer 1 Comments

1. Summary

We sincerely appreciate the time and effort dedicated to reviewing our manuscript. Below, we provide detailed responses to each comment and outline the revisions made in the resubmitted manuscript. All modifications are highlighted in track changes for ease of reference. We remain open to further adjustments to meet the journal’s standards.

2. Point-by-point response to Comments and Suggestions for Authors

Comments 1: The section on therapeutic interventions is comprehensive but needs to be more detailed, particularly in terms of analysing preclinical and clinical studies. In particular, a more detailed examination of ferroptosis inhibitors, NLRP3 antagonists and other treatments would strengthen the discussion. The inclusion of data from clinical trials or case studies would further validate the proposed pathways and increase their credibility.

Response 1:

Thank you for this insightful suggestion. Revised sections detail ferroptosis (Page 4, lines 130–141), cuproptosis (Page 7, lines 253–267), autophagy (Page 8, lines 328–352), and pyroptosis (Page 14, lines 442–456) inhibitors, linking preclinical mechanisms to clinical potential in male infertility.We acknowledge the limited availability of clinical studies directly linking ferroptosis/cuproptosis to male infertility. To address this, we expanded our analysis as follows:

1. Clinical Relevance in Related Fields:

·       Added findings from cancer studies to infer translational potential. For example:

"Some clinical studies suggests that a potent copper chelator Bis-choline tetrathio-molybdate  (TTM) reduced intestinal copper uptake to a clinically significant degree in Wilsons disease patients and revealed well tolerated in patients with breast cancer (Page 7, lines 256258)."

·       Incorporated bioinformatics evidence:

"Bioinformatic analyses reveal that elevated FDX1 expression, a cuproptosis marker, correlates with poor prognosis in glioma and gastric cancer (Page 7, lines 261263)."

2. Antioxidant Clinical Trials:

·       Supplemented with randomized controlled trials (RCTs) on antioxidants:

"Related randomised controlled trials suggest antioxidant supplementation may improve live birth rates in subfertile males (Page 3, lines 119-120)."

3. NLRP3 Inhibitors:

·       Highlighted preclinical evidence due to lack of clinical trials:

"Male infertility due to an inflammatory response triggered by Coronavirus disease 2019 is attenuated by NLRP3 inflammasome inhibitors, such as MCC950 and parthenolide. Three FDA-approved drugs (ChEMBL 4297185, ChEMBL 1201749, ChEMBL 1200545) selected by virtual screening also treat both COVID-19 and male varicocele by inhibiting the NLRP3 inflammasome. (Page 14, lines 449–453)."

4.Autophagy-related intervention supplementation

·       Complemented respectively interventions for male infertility due to lifestyle and environmental factors:

“Sancai Lianmei granules improve sperm density and viability in obese mice by inhibiting excessive autophagy.” (Page 8, lines 329–331).

“Berberine nanostructured biloalbuminosomes (BBR-BILS) and M1 nanostructured biloalbuminosomes (M1-BILS) ameliorate Cd- and Pb-induced testicular and prostate damage in rats by attenuating oxidative stress-induced excessive autophagy.” (Page 8, lines 340–342).

Ginkgo biloba extract attenuates insecticide deltamethrin-induced testicular damage in mice by upregulating SKP2, a key regulator of Beclin1-independent autophagy, and inhibiting Beclin1 activation to regulate autophagy.” (Page 8, lines 344–346).

·       Added clinical evidence:

“In addition, clinical study have found that infertile oligoasthenoteratozoospermic men associated with varicocele have significantly higher cystatin C levels in semen than healthy males. And six months after surgery, the seminal cystatin C levels were sig-nificantly lower than pre - operative levels. This implied that cystatin C may ameliorate male infertility due to varicocele by modulating autophagy.” (Page 8, lines 346–350).

Comments 2:In addition, the clarity of the illustrations, especially Figure 1, should be improved. The current layout makes it difficult to distinguish the individual components. Increasing the font and improving the general clarity would make it easier to interpret the illustrations.

Response 2: Thank you for pointing this out. We agree with this comment. Therefore, we haveincreased the font and improved the general clarity.

 Comments 3:In addition, the section "Disulfide death" in Figure 3 should be removed as it is not mentioned in the text.

Response 3: Agreed. The "Disulfide death" panel in Figure 3 has been removed to align the figure with the manuscript content.

 Comments 4: Line 124 - it would be useful to add here that Ferrostatin-1 is a ferroptosis inhibitor

Response 4: We have revised the text to explicitly define Ferrostatin-1:

"Ferrostatin-1 (Fer-1) and deferoxamine (DFO), two well-characterized ferroptosis inhibitors, mitigate oxidative stress and lipid peroxidation in testicular cells. (Page 4, lines 133–136)."

 3. Response to Comments on the Quality of English Language

 Point 1: Line 64 – systematically instead systemativally

Response 1: Corrected to:

"This review systematically examines the interplay between cell death modalities and male reproductive dysfunction..." (Page 2, line 63).

 Point 2: Line 170 – transferrin abbreviation here, not in line 171

Response 2: Revised to:

"Critical factors, including transferrin (TF) and Nrf2, mediate ferroptosis in testicular cells (Page 5, line 175)."

Point 3: Line 474 – cuproptosis vs cuproposis

Response 3: Amended throughout the manuscript, e.g.:

"Common mechanisms in ferroptosis and cuproptosis include oxidative stress and mitochondrial dysfunction (Pages 17, lines 490–500)."

 4. Additional clarifications

No further clarifications are required at this stage. We are pleased to address any additional concerns the editor or reviewers may have.

Reviewer 2 Report

Comments and Suggestions for Authors

The article “Interplay of Ferroptosis, Cuproptosis, Autophagy and Pyroptosis in Male Infertility: Molecular Crossroads and Therapeutic Opportunities” by Cai et al., is intended to revise the role of ferroptosis, cuproptosis, autophagy, and pyroptosis on male infertility; however, some issues should be pointed out:

1.- Line 11. Please remove from this sentence the definition of autophagy

2.- Line 47. Male infertility is multifactorial condition, authors may add more information about this factor in this sentence, using proper references.

3.- Figure 1. This figure lacks legends, it looks somehow incomplete, e.g. authors may add the legend of the testicle and its parts

4.- Line 88. Please define what is varicocele

5.- Line 139. Please add EC numbers of superoxide dismutase

6.- Line 152. Please add EC numbers Glutathione synthase

7.- Line 266. Please add a proper and complete definition of autophagy in this sentence

8.- Line 276. Please define ATG7 and all autophagy-related proteins when first mentioned in this section

9.- Line 368. Authors are excluding the role of rapamycin in this section, since is already reported that the use or this drug alters the spermatogenesis targeting mTOR. Authors may add this information or explain why it is excluded

10.- Line 371. Please define pyroptosis

11.- Line 513. This section should be called as conclusion instead of summary

Author Response

Response to Reviewer 2 Comments

1. Summary

We sincerely appreciate the thorough review of our manuscript and the constructive feedback provided. Below, we address each comment in detail and outline the revisions implemented in the resubmitted manuscript. All changes are highlighted in track changes for clarity.

2. Point-by-point response to Comments and Suggestions for Authors

Comments 1: Line 11. Please remove from this sentence the definition of autophagy

Response 1: Thank you for this observation. The definition of autophagy has been removed from Line 11 to maintain conciseness.

Comments 2: Line 47. Male infertility is multifactorial condition, authors may add more information about this factor in this sentence, using proper references

Response 2: We have revised the section to include a comprehensive list of contributing factors, supported by recent references:

"Male infertility is multifactorial, influenced by environmental pollutants (e.g., endocrine disruptors), genetic anomalies (e.g., Y-chromosome microdeletions), congenital abnormalities (e.g., cryptorchidism), infections (e.g., SARS-CoV-2-induced paediatric acute epididymo-orchitis), lifestyle factors (e.g., smoking, obesity), and sociodemographic trends such as delayed parenthood (Page 2, lines44-48)."

 Comments 3:  Figure 1. This figure lacks legends, it looks somehow incomplete, e.g. authors may add the legend of the testicle and its parts

Response 3: The figure legend has been expanded for clarity:

"Figure 1. Modes of cell death in the human testis. The schematic illustrates distinct forms of cell death within testicular compartments. Color coding: blue (seminiferous tubules, housing germs cells), red (Sertoli cells), and yellow (Leydig cells). Specific cell death modalities are annotated as follows:……”(Page 2, Figure 1 legend)."

Comments 4: Line 88. Please define what is varicocele

Response 4: The definition has been refined for precision:

"Varicocele (VC), a prevalent cause of male infertility, is characterized by abnormal dilation of the pampiniform venous plexus, leading to impaired thermoregulation, oxidative stress (e.g., reduced glutathione peroxidase activity), and compromised spermatogenesis (Page 3, lines89-92)

Comments 5:Line 139. Please add EC numbers of superoxide dismutase

Response 5: Agree. We have, accordingly added EC numbers. The adderd content is:

In the presence of mitochondrial superoxide dismutase (SOD, EC 1.15.1.1)”(page5,line145).

Comments 6:Line 152. Please add EC numbers Glutathione synthase

Response 6: Agree. We have, accordingly added EC numbers. The adderd content is

Glutathione synthase (GSS, EC 6.3.2.3) catalyzes the final step in GSH synthesis”(page5,line156).

Comments 7: Line 266. Please add a proper and complete definition of autophagy in this sentence

Response 7: The definition has been revised for accuracy:

"Autophagy is a conserved lysosome-dependent process wherein cytoplasmic components (e.g., damaged organelles) are sequestered within double-membraned autophagosomes and degraded to maintain cellular homeostasis (Page 7, lines 269-271)."

Comments 8: Line 276. Please define ATG7 and all autophagy-related proteins when first mentioned in this section

Response 8: Thank you for pointing this out. We agree with this comment. Therefore, We have added a paragraph in this section to introduce the autophagy related proteins mentioned. The modified content is:

“In autophagy, multiple key proteins work together to ensure autophagosomes formation, expansion, and fusion with lysosomes. Beclin-1(BECN1) initiates autophagosome formation by interacting with the phosphatidylinositol 3-kinase complex. Autophagy related 7 (ATG7), an e1 like activating enzyme, promotes au-tophagosome membrane expansion by facilitating autophagy related 5 (ATG5) and ATG12(autophagy-related protein 12) binding. ATG5 and ATG12 form a complex that binds with autophagy related 16 like 1 (ATG16L1), further extending and closing the autophagosome membrane. Microtubule-associated protein 1 light chain 3 (LC3 proteins), markers of autophagosome membranes, convert from soluble LC3-I to membrane-bound LC3-II, integrating LC3-II into au-tophagosome membranes to aid their formation and expansion. Lysosomal associated membrane protein 2 (LAMP2), a crucial lyso-somal membrane protein, is involved in the fusion of autophagosomes with lysosomes. These proteins collaborate in autophagy to maintain cell homeostasis.GABA type A receptor-associated protein(GABARAP) is a ubiquitin like protein belonging to the ATG8 family, involved in autophagosome formation, expansion, and fusion with lysosomes.PTEN-induced putative kinase 1 ( PINK1), a serine/threonine kinase,is a sensor of mitochondrial damage and mitochondrial autophagy (mitophagy). (pages7, lines271-286).

Comments 9: Line 368. Authors are excluding the role of rapamycin in this section, since is already reported that the use or this drug alters the spermatogenesis targeting mTOR. Authors may add this information or explain why it is excluded

Response 9: Thank you for pointing this out. We agree with this comment. Therefore, We have cited references to demonstrate that mTOR can improve spermatogenesis. The modified content is:

“mTOR acts as a core factor regulating spermatogenesis, and the use of mTOR inhibitors such as rapamycin can improve spermatogenesis by activating autophagy in the presence of autophagy deficiency (page9, lines379-381).

Comments 10: Line 371. Please define pyroptosis

Response 10: The definition has been relocated and expanded:

"Pyroptosis, a pro-inflammatory programmed cell death driven by gasdermin family proteins (e.g., GSDMD), is characterized by plasma membrane pore formation, cytokine release (e.g., IL-1β), and osmotic lysis. It plays dual roles in testicular immune defense and pathological inflammation (Page 10, lines 395-398)."

3. Response to Comments on the Quality of English Language

Point 1: Line 513. This section should be called as conclusion instead of summary

Response 1: The section heading has been updated to:

"5. Conclusion" (Page 16, line528).

4. Additional clarifications

No further clarifications are required at this stage. We welcome any additional feedback to enhance the manuscript’s rigor and clarity.

Reviewer 3 Report

Comments and Suggestions for Authors

Dear authors, although it is largely a complete review, there are some aspects that should be improved. 

The text is dense with biochemical markers, gene names, and oxidative stress indicators (e.g., GPX4, NRF2, FDX1, SLC7A11), which might be difficult for a non-specialist audience. Provide brief explanations or context.

Certain ideas are repeated with slight variations. Example:"Ferrostatin-1 (Fer-1) and deferoxamine (DFO) have been shown to mitigate oxidative stress and lipid peroxidation, thus improving sperm quality and fertility outcomes in animal models."  The next sentence reiterates how Fer-1 helps sperm motility, which is already implied.

While biological mechanisms are well-covered, the clinical significance and diagnostic applications of cuproptosis markers in human male infertility are briefly mentioned but not elaborated. Add more discussion on how these findings can be translated into clinical practice.

Author Response

Response to Reviewer 3 Comments

1. Summary

We extend our sincere gratitude for the thorough evaluation of our manuscript and the constructive feedback. Below, we address each comment in detail, outlining the revisions implemented to enhance clarity, reduce redundancy, and strengthen clinical relevance. All modifications are highlighted in the resubmitted manuscript.

2. Point-by-point response to Comments and Suggestions for Authors

Comments 1: The text is dense with biochemical markers, gene names, and oxidative stress indicators (e.g., GPX4, NRF2, FDX1, SLC7A11), which might be difficult for a non-specialist audience. Provide brief explanations or context.

Response 1:We appreciate this suggestion and have integrated concise definitions and functional context for key terms: GPX4 (Glutathione peroxidase 4): A selenium-dependent antioxidant enzyme critical for neutralizing lipid peroxides, thereby preventing ferroptosis (Page 3, lines 80-82). NRF2 (Nuclear factor erythroid 2-related factor 2): A master regulator of antioxidant defense, activating GPX4 to counteract oxidative stress. Beyond ferroptosis, Nrf2 dysregulation is implicated in neurodegenerative diseases, cancer, and metabolic disorders, underscoring its broad pathophysiological relevance (Page 3, lines 102-105). FDX1 (Ferredoxin 1): A mitochondrial iron-sulfur protein implicated in cuproptosis, linking copper toxicity to lipoylated TCA cycle enzyme aggregation (Page3, lines112-113). SLC7A11(Solute carrier family 7 member 11): A cystine/glutamate antiporter essential for glutathione synthesis, acting as a gatekeeper of ferroptosis susceptibility (Page3, lines 84-85).

Comments 2: Certain ideas are repeated with slight variations. Example:"Ferrostatin-1 (Fer-1) and deferoxamine (DFO) have been shown to mitigate oxidative stress and lipid peroxidation, thus improving sperm quality and fertility outcomes in animal models."  The next sentence reiterates how Fer-1 helps sperm motility, which is already implied.

Response 2: Thank you for highlighting this redundancy. The text has been consolidated into a single, cohesive paragraph:

"Ferrostatin-1 (Fer-1) and deferoxamine (DFO), two well-characterized ferroptosis inhibitors, mitigate oxidative stress and lipid peroxidation in testicular cells. In busulfan-induced oligospermia models, Fer-1 restored sperm quality by upregulating GPX4 and modulating NRF2 signaling, underscoring its therapeutic potential in male infertility (Page 4, lines 131–134)."

Comments 3: While biological mechanisms are well-covered, the clinical significance and diagnostic applications of cuproptosis markers in human male infertility are briefly mentioned but not elaborated. Add more discussion on how these findings can be translated into clinical practice.

Response 3: While direct clinical studies on cuproptosis in male infertility remain limited, we have strengthened the translational discussion by integrating findings from oncology and proposing hypotheses:

"Bioinformatic analyses reveal that elevated FDX1 expression correlates with poor prognosis in glioma and gastric cancer, suggesting its role in copper-driven cell death (Page 7, lines 260–262). Translating these findings to male infertility, we hypothesize that FDX1 overexpression in testicular cells may exacerbate copper toxicity, impairing spermatogenesis. Future studies should explore urinary/serum copper levels and FDX1 activity in infertile males, potentially establishing FDX1 as a diagnostic biomarker or therapeutic target (Page 17, lines 500–504)."

3. Additional clarifications

No further clarifications are required at this stage. We welcome additional feedback to refine the manuscript further.

Reviewer 4 Report

Comments and Suggestions for Authors

In this review, authors synthesized current molecular insights into cell death pathways implicated in male infertility, highlighting their interplay and translational potential for restoring spermatogenic function.

The manuscript is interesting, well illustrated and quite well written. However, there are several points that deserve to be improved. In particular:

Lines 45-49: it deserves to be added that male fertility and semen quality can also be impaired by infections (see PMID: 35114008)

Line 80-81: Heat shock transcription factor 1  (HSF1), Kelch domain containing 3 (KLHDC3) 

Lines 100-103: although authors mentioned NRF2, they did not explain the function of this transcription factor. Since this is a review article, authors shoud give a general overview of the topics treated. To this aim, the multifaceted function of this transcription factor deserves to be highlighted since it plays a key role in the antioxidant response and in the onset and progression of several diseases (see PMID: 39769005). 

Line 147: "... subsequent ferroptosis[18, 20] [25] [43]." references must be shown according to the journal style

Uniform Nrf2 throughout the text (Nrf2  or NRF2)

Abbreviations must be written in full lenght when mentioned for the first time

Authous should state which program they used to make the figures under each figure legend

Text must be formatted uniformly throughout the manuscript

An accurate revision of typing errors is recommended

Author Response

Response to Reviewer 4 Comments

1. Summary

We extend our sincere gratitude for the thorough evaluation of our manuscript and the constructive feedback. Below, we address each comment in detail, outlining the revisions implemented to enhance clarity, reduce redundancy, and strengthen clinical relevance. All modifications are highlighted in the resubmitted manuscript.

 2. Point-by-point response to Comments and Suggestions for Authors

Comments 1: Lines 45-49: it deserves to be added that male fertility and semen quality can also be impaired by infections (see PMID: 35114008)

Response 1: Thank you for pointing this out. We agree with this comment. Therefore, we have supplemented relevant factors as suggested. The added content is

"Male infertility is multifactorial, influenced by environmental pollutants (e.g., endocrine disruptors), genetic anomalies (e.g., Y-chromosome microdeletions), congenital abnormalities (e.g., cryptorchidism), infections (e.g., SARS-CoV-2-induced paediatric acute epididymo-orchitis), lifestyle factors (e.g., smoking, obesity), and sociodemographic trends such as delayed parenthood (Page 2, lines44-48)."

Comments 2: Line 80-81: Heat shock transcription factor 1 (HSF1), Kelch domain containing 3 (KLHDC3)

Response 2: Thank you for pointing this out. We agree with this comment. Therefore, we have supplemented them:

“Heat shock transcription factor 1 (HSF1), and Kelch domain containing 3 (KLHDC3) may be involved in ferroptosis in non-obstructive azoospermia (NOA)”(Page3,lines80-83).

Comments 3: Lines 100-103: although authors mentioned NRF2, they did not explain the function of this transcription factor. Since this is a review article, authors shoud give a general overview of the topics treated. To this aim, the multifaceted function of this transcription factor deserves to be highlighted since it plays a key role in the antioxidant response and in the onset and progression of several diseases (see PMID: 39769005)

Response 3: We expanded the discussion on Nrf2 to emphasize its multifunctional role:

" Nrf2 has been identified as a master regulator of antioxidant defense, activating GPX4 to counteract oxidative stress. Beyond ferroptosis, Nrf2 dysregulation is implicated in neurodegenerative diseases, cancer, and metabolic dis-orders, underscoring its broad pathophysiological relevance (Page 3, lines 102–105)."

Comments 4:Line 147: "... subsequent ferroptosis[18, 20] [25] [43]." references must be shown according to the journal style

Response 4: References have been standardized to sequential numbering:

"... subsequent ferroptosis[22, 24, 29, 51]" (Page 5, line 152).

Comments 5:Uniform Nrf2 throughout the text (Nrf2  or NRF2)

Response 5: We carefully read through the entire article and reviewed and modified the expression patterns of the corresponding genes and proteins according to the differences in the references.

Comments 6:Abbreviations must be written in full lenght when mentioned for the first time

Response 6: We read through the entire article carefully, and when the abbreviation first appeared, the corresponding full length was added.

Comments 7: Authous should state which program they used to make the figures under each figure legend

Response 7: Figure legends now specify the software:

"The diagram was drawn using Adobe Illustrator." (Page 2, Figure 3 legend; Page 4, Figure 2 legend; Page 18, Figure 3 legend).

Comments 8 : Text must be formatted uniformly throughout the manuscript

Response 8:  Throughout the manuscript, the text is formatted in a uniform body font: Times New Roman.

Comments 9 : An accurate revision of typing errors is recommended

Response 9: We carefully read through the entire article and revised the relative typing errors.

3. Additional clarifications

No further clarifications are required at this stage. We are pleased to address any additional concerns the editor or reviewers may have.

Round 2

Reviewer 1 Report

Comments and Suggestions for Authors

I have reviewed your article and am pleased to inform you that you have successfully met all the required criteria. The content is comprehensive, well organised and meets publication standards. After careful consideration, I am of the opinion that the article is sufficiently strong and suitable for publication.

Reviewer 2 Report

Comments and Suggestions for Authors

No further comments

Reviewer 4 Report

Comments and Suggestions for Authors

the manuscript has been significantly improved and can be accepted in the present form